# Determinants of personal vaccination hesitancy before and after the mid-2021 COVID-19 outbreak in Taiwan

**Hsuan-Wei Lee** [1]*, **Cheng-Han Leng**[2], **Ta-Chien Chan** [3]

**1** Institute of Sociology, Academia Sinica, Taipei City, Taiwan, **2** Department of Psychology, National Taiwan University, Taipei City, Taiwan, **3** Research Center for Humanities and Social Sciences, Academia Sinica, Taipei City, Taiwan

* hwwaynelee@gate.sinica.edu.tw

## Abstract

### Background

Using a 10 week nationwide online survey performed during a time period containing the time ahead, the start, and the peak of a COVID-19 outbreak in Taiwan, we investigated aspects that could affect participants' vaccination intentions.

### Methods

From March to May 2021, we surveyed 1,773 people in Taiwan, aged from 20 to 75 years, to determine potential acceptance rates and factors influencing the acceptance of a COVID-19 vaccine. We used an ordinal logistic regression with a backward selection method to identify factors that affected vaccination intention.

### Results

Several factors could increase individuals' vaccination intentions including: being male, older, with an openness personality, having a better quality of life in the physical health domain, having better knowledge and personal health behavior, having more trust in the government, and being worried about misinformation. Perceived risks played a crucial role in the vaccine decision-making process. When the pandemic intensified, people's vaccination intentions increased significantly.

### Conclusion

The findings of the present study could highlight individuals' vaccination attitudes and provide governments with an empirical and dynamic base to design tailored strategies to increase vaccination rates.

**Data Availability Statement:** All relevant data are within the paper.

**Funding:** This research was funded by the Academia Sinica, Taiwan https://www.sinica.edu.tw/en (AS-IRB-HS07-109104 and AS-IRB-HS07-

109090) for the first author. The funders had no role in study design, data collection and analysis, decision to publish, or preparation of the manuscript.

**Competing interests:** The authors have declared that no competing interests exist.

## Introduction

As SARS-CoV-2 infections have surged around the world [1], countries have been eager to reach higher vaccination rates among their citizens to achieve herd immunity and prevent the further spread of the pandemic. Some governments made COVID-19 vaccinations mandatory for health practitioners and other high-risk groups [2], while other countries like the United States and France debated the vaccine mandate. Therefore, understanding the factors that affect people's vaccination intentions is crucial for governments to effectively increase the vaccination rate in their countries.

Multiple studies have been conducted on vaccine intention in different countries such as Australia [3, 4], Canada [5], China [6], Czechia [7, 8], France [9], Germany [10], Greece [11], Hong Kong [12, 13], India [14], Indonesia [15], Japan [16, 17], Malaysia [18], New Zealand [19], Portugal [20], Saudi Arabia [21, 22], Slovenia [23], the United Kingdom [24–27], the Caribbean region [28], the United States [29–33], and Taiwan [34, 35]. There are also cross-country surveys [36–43]. Most of these studies offered public health perspectives and investigated major demographic factors that influenced people's vaccination intention during certain snapshots of time. Among various vaccine intention studies, one study conducted a multinational survey in June 2020 involving 13,426 people from 19 countries, among them, 71.5% of participants reported that they would be very or somewhat likely to take a COVID-19 vaccine [42]. Respondents reporting higher levels of trust in information from government sources were more likely to accept a vaccine and to take their employer's advice to do so. A recent study performed a systematic literature search of PubMed and Web of Science before July 2021, and showed a variety of factors that could lead to vaccine hesitancy, including having a negative perception of vaccine efficacy, safety, convenience, and price [44].

Compared to other countries, Taiwan had remarkable success in containing the COVID-19 epidemic [45–48], recording zero local spread cases from April to December 2020. While many countries around the world suffered from the COVID-19 pandemic in 2020, due to no local epidemic, the vaccine acceptance rate in Taiwan is much lower than in neighboring countries at the beginning of vaccination campaign [43]. As of April 2021, a study had performed an online survey over 18 days to collect a sample of 1,100 responses in Taiwan [35]. The authors found that certain demographic characteristics including being male and psychological factors such as the belief in the artificial origin of the virus could suppress people's COVID-19 vaccination intention in Taiwan. Though insightful, one limitation of this study was its timeliness. This study was conducted in early April 2021, when Taiwan's daily number of infected cases was extremely low (0.14 daily new confirmed COVID-19 cases per million people [49]). However, in May 2021, the COVID-free normality enjoyed by Taiwanese for almost a year ended. The government issued a Level 3 pandemic alert on May 15, 2021 (1.75 daily new confirmed COVID-19 cases per million people [49]), to limit the spread of the virus. The outbreak soon reached its peak in late May. During the 2021 outbreak, the Taipei metropolitan area became an epicenter for infections. During the outbreak, the vast majority was unvaccinated then, only less than 1% of the population was vaccinated [50]. Noticeably, there were insufficient vaccines for most residents in Taiwan causing the population to feel stressed and anxious [51].

We assumed that although the different variants of SARS-CoV-2 virus were rampant around the world, the lack of infections in Taiwan meant that people's vaccine intentions remained low. By contrast, once threats were imminent, such as due to a local outbreak, people's perceived risk and willingness to be vaccinated would increase dramatically. This study aimed to explore people's motivations to be vaccinated in association with factors such as their demographic characteristics, psychological perspectives, health-related behavior, political

attitudes, and most importantly, the COVID-related risk factors including people's locations, quarantine experiences, and the number of daily new confirmed cases in Taiwan. Considering all these factors, we aimed to carry out holistic investigation of why people either do or do not want to be vaccinated. Therefore, the present study has the following three key research questions (RQs): **RQ1**: What proportion of people would accept a vaccine for COVID-19? **RQ2**: What sociodemographic factors, psychological factors, and health and political attitudes are associated with the intention to accept a future vaccine for COVID-19? **RQ3**: How do COVID-related risk factors and potential threats affect one's vaccine intention?

## Materials and methods

To investigate which factors affect people's willingness to get vaccinated, we performed a 10-week online nationwide survey in Taiwan. Fig 1 illustrates Taiwan's mid-2021 COVID-19 outbreak and our survey period [49]. Our survey period covered the early development and the peak of the mid-2021 outbreak. During this period, the population of Taiwan experienced a substantial change in their attitude toward COVID-19.

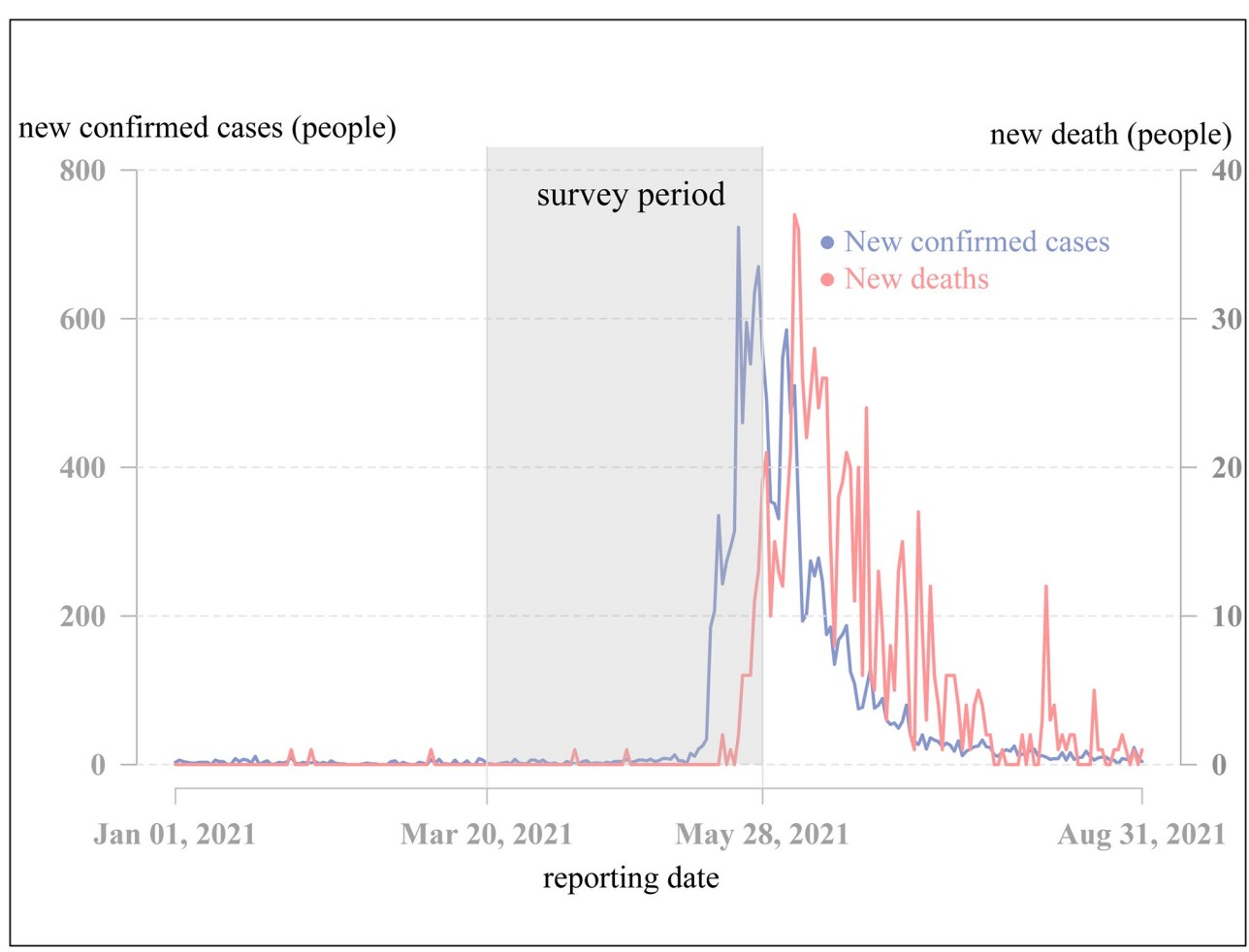

**Fig 1. The timeline and casualties of Taiwan's mid-2021 COVID-19 outbreak and our survey period.**

## Design

We conducted a cross-sectional survey, from March 20 to May 28, 2021. Participants were recruited via multiple social media platforms and were directed to our website–Social Distancing Survey–where they could complete the survey with ethical approval (reference: AS-IRB-HS07-109104). Participants entered our website after confirming the electronic informed consent form, which was printed on the front page.

## Questionnaire

Our survey consisted of five main parts: the participants needed to provide their (i) basic demographic characteristics, (ii) psychological perspectives, (iii) public health knowledge and personal health behavior, and (iv) attitude towards the government among different aspects, and (v) any COVID-related risk factors.

In the psychological part of the survey, participants' quality of life (QoL) was measured by the WHOQOL-BREF [52]. The WHOQOL-BREF contains four domains: physical health (seven items), psychological health (six items), social relationships (three items), and environment (eight items). Each item followed a five-point Likert scale ranging from low to high QoL. Moreover, the personality traits consist of five factors, namely, openness to experience, conscientiousness, extraversion, agreeableness, and neuroticism [53]. These traits were measured by the Big-Five Inventory [54], and each trait was measured by two items. The scale that was used followed the five-point Likert scale.

To understand if the participants had enough public health knowledge and had been practicing good health habits since the global outbreak of COVID-19, we asked how frequently they came into physical contact with other people and entered crowded places. Moreover, we asked if the participants' would tell their doctors about their sickness and keep social distance if they feel ill, to measure their public health knowledge and attitudes.

We also measured a wide range of the participants' political attitudes toward the Taiwan government. We asked if the participants were satisfied with the government's performance when dealing with the COVID-19 outbreak; determining if the participant agreed that (a) the government should restrict individual freedom in order to control the COVID-19 outbreak, (b) the government should track how people move around in order to control the outbreak, (c) the government should release personal information of COVID-19 patients for the sake of enhancing people's understanding of the epidemic situation, (d) the government should provide financial aid due to the economic crisis caused by the outbreak. We also asked the participants (e) how trustworthy they found the information provided by their government about the development of the epidemic, and (f) how worried they were about the negative impact of misinformation related to epidemic prevention on society.

Lastly, we investigated the risk factors the participants had in relation to the COVID-19 pandemic. Four additional factors were explored in this category: (a) if the participants were living in Taipei or not, (b) if they had quarantine experience related to COVID-19, (c) whether the participants have had a COVID-19 test, (d) whether the time the participants took the survey was during the national COVID-19 level three alert, and (e) the previous daily new confirmed cases.

## Outcome measure

To measure vaccination intention, participants were asked to respond to the item "when a COVID-19 vaccination becomes available to [them]" with the Likert type choices: "1: I won't take it", "2: Maybe, maybe not", and "3: I will take it". Meanwhile, participants were also asked to provide the reasons for their answers (via multiple choice) "that will affect their incentive to

take the COVID-19 vaccine" with the choices of "the physical impact of the vaccine", "the mental impact of the vaccine", "the efficacy of the vaccine", "the price of the vaccine", "the vaccination process", and "other reason", each reason was recorded as a dummy variable.

### Analysis

The present study conducted all analyses using R software in version 4.1.1 and used an ordinal logistic regression to build the model with the package "ordinal" [55]. The dependent variable was the strength of the motivation for a vaccination, and the independent variables were the five categories of variables mentioned in the Questionnaire subsection, in addition to the reasons influencing their motivation, as demonstrated in Table 1. Due to the number of variables included, we used a backward selection approach to select the variables with a significance level of.05.

## Results

As demonstrated in Table 2, about half of our participants lived in Taipei (57.53%), two thirds of which were female (67.75%) and aged less than thirty years old (38.04%). Most of the participants were single (83.14%), had higher education (97.94%), and had household income between 1,700 and 3,399 or between 3,400 and 6,799 USD a month. The majority of the participants submitted their responses during the level three alert (76.67%), have not tested for the COVID-19 virus (93.43%), and did not have quarantine experience (89.61%).

### RQ1: What proportion of people would accept a vaccine for COVID-19?

As demonstrated in Table 3, about half of our participants were willing to vaccinate (52.99 −54.51%), which did not differ according to their living place. Also, roughly 40% of the participants might take the vaccination (41.04−40.78%). Most of the participants cared about the risk of the physical impact of the vaccine (80.21−79.99%) and the efficacy of the vaccination (70.65 −41.47%). Although it was free to vaccinate in Taiwan, about 40% of the participants remained focused on the price of the vaccine (34.79−41.47%), which might relate to the notion of being vaccinated abroad. About one-third of the participants were worried about the risk of any psychological impact of the vaccine (21.27−21.78%) and the risk of the vaccination process (18.99 −20.20%). Few of the participants were worried about other reasons (6.47−8.10%).

### RQ2: What sociodemographic factors, psychological factors, and health and political attitudes are associated with the intention to accept a future vaccine for COVID-19

As shown in Table 4, among the demographic factors, the willingness of men to vaccinate was 1.48 times that of women, and that there was 1.49 times as many participants over the age of 30 as opposed to below the age of 30.

Among the psychological perspectives, people who scored one more score in the personality trait of openness would increase their vaccination willingness with a multiple of 1.07. In other words, the more inventive/curious the participants were towards new experiences, the more likely they were to want to get vaccinated. Besides, in the QoL section, those who scored one more score in the physical health domain would like to get vaccinated with a multiple of 1.08. Nevertheless, higher social relationships lead to a higher chance of vaccination hesitancy (motivation increased by a multiple of 0.94 per unit).

**Table 1. All candidate variables in the full model.**

| | |
|---|---|
| a-1 | gender–male (with the reference of female) |
| a-2 | aged over 30 years old (with the reference of aged below 30 years old) |
| a-3 | marital status–single (with the reference of married) |
| a-4 | education |
| a-5 | living place–Taipei (with the reference of outside Taipei) |
| a-6 | household income |
| b-1 | during the level three alert (with the reference of before the level three alert) |
| b-2 | test result–negative (with the reference of untested) |
| b-3 | quarantine condition–never-quarantined (with a reference of ever-quarantined) |
| b-4 | yesterday's new confirmed cases |
| c-1 | openness to experience |
| c-2 | conscientiousness |
| c-3 | extraversion |
| c-4 | agreeableness |
| c-5 | neuroticism |
| d-1 | physical health |
| d-2 | psychological health |
| d-3 | social relationships |
| d-4 | environment |
| e | daily contact on average |
| f | the sufficiency of requisite |
| g-1 | personal health behavior when making physical contact with others |
| g-2 | personal health behavior when in a crowded place |
| g-3 | personal health behavior of telling their doctor when feeling ill |
| g-4 | the necessity to wear a mask |
| h-1 | the satisfaction with the government |
| h-2 | the satisfaction with the World Health Organization (WHO) |
| h-3 | the agreement with the government to restrict personal movement |
| h-4 | the agreement with the government to track personal movement |
| h-5 | the agreement with the government to release patients' information |
| h-6 | the agreement with the financial aids provided by the government |
| h-7 | the trustworthiness of government information |
| h-8 | the worry of the negative impact of any misinformation |
| i | time spent on the internet |
| j-1 | considering the physical impact of the vaccination–yes (with a reference of no) |
| j-2 | considering the psychological impact of the vaccination–yes (with a reference of no) |
| j-3 | considering the vaccination price–yes (with a reference of no) |
| j-4 | considering the vaccination process–yes (with a reference of no) |
| j-5 | considering the other reasons for vaccination–yes (with a reference of no) |

In the participants' public health knowledge and personal health behavior section, those who usually go to crowded places (with a multiple of 1.28 per unit) or who were used to telling their doctor when feeling ill (with a multiple of 1.14 per unit) had more incentive to vaccinate.

Considering attitudes towards the government, people who strongly approved or supported that the government could track how people move around in order to control the outbreak (with a multiple of 1.14 per unit) and that the government should provide financial aid due to the economic crisis (with a multiple of 1.15 per unit), tended to have a higher vaccination willingness. In contrast, people who strongly disagreed with, the government releasing patients'

**Table 2. Demographic characteristics of the participants.**

| Characteristic | Total No. (%) | Residence No. (%) | | Population No. (%) | |
|---|---|---|---|---|---|
| | | Not Taipei | Taipei | Not Taipei | Taipei |
| Overall | 1,773 (100) | 753 (42.47) | 1,020 (57.53) | 16,593,952 (70.77) | 6,967,274 (29.23) |
| Gender | | | | | |
| Male | 555 (31.30) | 226 (30.01) | 329 (32.25) | 11,616,647 (49.61) | |
| Female | 1,218 (68.70) | 527 (69.99) | 691 (67.75) | 11,835,190 (50.39) | |
| Age | | | | | |
| < 30 years old | 1,086 (61.25) | 454 (60.29) | 632 (61.96) | 7,210,318 (30.80) | |
| ≥ 30 years old | 687 (38.74) | 299 (39.71) | 388 (38.04) | 16,350,918 (69.20) | |
| Marital status | | | | | |
| Married | 336 (18.95) | 164 (21.78) | 172 (16.86) | - | |
| Single | 1,437 (81.06) | 589 (78.22) | 848 (83.14) | - | |
| Education | | | | | |
| Education | 66 (3.72) | 45 (5.98) | 21 (2.06) | - | |
| Higher Education | 1,707 (96.28) | 708 (94.02) | 999 (97.94) | - | |
| Household income | | | | | |
| ≤ 329 USD/month | 60 (3.38) | 32 (4.25) | 28 (2.75) | - | |
| 330-989 USD/month | 133 (7.50) | 68 (9.03) | 65 (6.37) | - | |
| 990-1,699 USD/month | 318 (17.94) | 132 (17.53) | 186 (18.24) | - | |
| 1,700-3,399 USD/month | 611 (34.46) | 278 (36.92) | 333 (32.65) | - | |
| 3,400-6,799 USD/month | 531 (29.95) | 203 (26.96) | 328 (32.16) | - | |
| ≥ 6,800 USD/month | 120 (6.77) | 40 (5.31) | 80 (7.84) | - | |
| Level three alert | | | | | |
| Before | 363 (20.47) | 125 (16.60) | 238 (23.33) | - | |
| During | 1,410 (79.53) | 628 (83.40) | 782 (76.67) | - | |
| COVID-19 test result | | | | | |
| Not tested | 1,668 (94.08) | 715 (94.95) | 953 (93.43) | - | |
| Negative | 105 (5.92) | 38 (5.05) | 67 (6.57) | - | |
| Quarantine experience | | | | | |
| Ever-quarantined | 168 (9.48) | 62 (8.23) | 106 (10.39) | - | |
| Never-quarantined | 1,605 (90.52) | 691 (91.77) | 914 (89.61) | - | |

USD: United States Dollar

information (with a multiple of 0.87 per unit) were more willing to vaccinate. Moreover, those who thought the information provided by their government about the development of the epidemic was trustworthy (with a multiple of 1.35 per unit) or if they were worried about the negative impact of misinformation related to epidemic prevention on society (with a multiple of 1.14 per unit), tended to have a higher preference to vaccinate.

## RQ3: How do COVID-related risk factors and potential threats affect one's vaccine intention?

When the Taiwanese government issued a COVID-19 level three alert, the willingness of participants to vaccinate increased by a multiple of 2.8. The previous daily new confirmed cases was statistically insignificant, and we found that its effect was offset by the COVID-19 level three alert. Those who tested negative for the COVID-19 virus were more willing to vaccinate with a multiple of 2. It is important to note that all the participants who had a COVID-19 test

**Table 3. Motivation for vaccination.**

| Characteristic | Total No. (%) | Residence No. (%) | |
|---|---|---|---|
| | | **Not Taipei** | **Taipei** |
| Overall | 1,773 (100) | 753 (42.47) | 1,020 (57.53) |
| Motivation for vaccination | | | |
| Will not take it | 93 (5.25) | 45 (5.98) | 48 (4.71) |
| Maybe, maybe not | 725 (40.89) | 309 (41.04) | 416 (40.78) |
| Will take it | 955 (53.86) | 399 (52.99) | 556 (54.51) |
| Reasons influencing the motivation | | | |
| Physical impact (yes) | 1,419 (80.03) | 604 (80.21) | 815 (79.9) |
| Psychological impact (yes) | 381 (21.49) | 164 (21.78) | 217 (21.27) |
| Efficacy (yes) | 1,271 (71.69) | 532 (70.65) | 739 (72.45) |
| Price (yes) | 685 (38.64) | 262 (34.79) | 423 (41.47) |
| Vaccination process (yes) | 349 (19.68) | 143 (18.99) | 206 (20.20) |
| Other (yes) | 127 (7.16) | 61 (8.10) | 66 (6.47) |

in the survey had negative results, therefore, the participants could have had a higher level of risk, pressuring them to take a COVID-19 test and thus had higher vaccination preference.

Lastly, among the reasons influencing the vaccine motivation, people who cared about the vaccination process (with a multiple of 0.71) or other reasons (with a multiple of 0.42) were unwilling to vaccinate. Considering the risk of taking the vaccination, participants worried about the vaccine's physical impact were more willing to take the vaccination than those who did not (with a multiple of 0.75).

## Discussion

The value of this study is highlighted by its time period, having been performed in a time window containing the time ahead, the start, and the peak of the most severe COVID-19 outbreak in Taiwan. For 253 days in 2020, Taiwan reported zero locally-transmitted cases of COVID-19. As we performed the survey from March to May 2021, people had experienced the substantial impacts of COVID-19 on their day-to-day life, changing their attitudes toward vaccinating. We examined various potential factors that could affect one's vaccination intentions and aimed to investigate the most important factors among various aspects that could lead to one's vaccinating decision. We clearly demonstrated that the participants' attitudes towards vaccinating had changed significantly as their risks of being infected increased. Moreover, there were not many available vaccines to choose from in Taiwan, and people did not have many discussions about multiple choices of vaccines. Therefore, the effect of vaccines' branding could be limited.

First of all, among all the sociodemographic factors we included, we found that only gender and age could be included in our final statistical model. Specifically, from our participants, men and people over 30 years old have higher vaccination intentions, which concurs with many previous studies in different countries [4, 5, 9, 12, 18, 27, 29, 31, 32] and in Taiwan by [35]. Furthermore, psychological indicators also affected vaccination intentions. Our study demonstrated that Individuals with higher openness are more likely to get vaccinated. As open individuals pursue new and unconventional ideas and experiences and tend to be flexible, inventive, and creative [56], they would be open to vaccination and thus have higher vaccination intentions.

When facing a pandemic, the results of our study showed that those who scored high for physical health tended to vaccinate. A possible explanation for this trend is that the habit for

**Table 4. Associated factors of the motivation for vaccination.**

| Variable | Estimate | Odds Ratio | 95% C.I. | Significance |
|---|---|---|---|---|
| Demographic factors | | | | |
| Male | 0.39 | 1.48 | (1.19, 1.84) | *** |
| Aged over 30 years old | 0.4 | 1.49 | (1.21, 1.84) | *** |
| Psychological perspectives | | | | |
| Openness to experience | 0.07 | 1.07 | (1.02, 1.14) | * |
| Physical health | 0.08 | 1.08 | (1.03, 1.14) | *** |
| Social Relationships | -0.06 | 0.94 | (0.90, 0.98) | ** |
| Public health knowledge and personal health behavior | | | | |
| Go to crowded places | 0.25 | 1.28 | (1.08, 1.51) | ** |
| Tell their doctor if feeling ill | 0.13 | 1.14 | (1.02, 1.27) | * |
| Attitude to the government | | | | |
| Movement tracking | 0.13 | 1.14 | (1.01, 1.27) | * |
| Release of patients' information | -0.14 | 0.87 | (0.79, 0.96) | ** |
| Financial aid | 0.14 | 1.15 | (1.00, 1.31) | ** |
| Trustworthy of government's information | 0.30 | 1.35 | (1.21, 1.52) | **** |
| Negative impact of misinformation | 0.13 | 1.14 | (1.00, 1.29) | * |
| COVID-related risk factors | | | | |
| Level three alert | 1.03 | 2.80 | (2.16, 3.65) | **** |
| Test negative | 0.69 | 2.00 | (1.29, 3.17) | ** |
| Reasons influencing the motivation for vaccination | | | | |
| Physical impact | -0.28 | 0.75 | (0.58, 0.97) | *** |
| Vaccination process | -0.34 | 0.71 | (0.56, 0.90) | ** |
| Other | -0.87 | 0.42 | (0.29, 0.61) | **** |

*: $p < .05$;

**: $p < .01$;

***: $p < .001$;

****: $p < .0001$.

individuals to maintain good physical health triggers them to vaccinate. Additionally, there was a negative indication of the social relationships dimension. We suggest that this dimension was a suppressor in our regression model and was not correlated to the willingness to be vaccinated (the correlation coefficient was only -0.0016 to the dependent variable) but was highly correlated with other variables and improved the performance of the model.

People who have better public health knowledge and personal health behavior tend to have a higher vaccination willingness [3, 29]. In the participants' public health knowledge and personal health behavior domain, we found that individuals who usually go to crowded places or who were accustomed to notifying their doctor when feeling ill had higher vaccination willingness. The effects of these factors were as expected. As many studies in health belief models suggested [29, 57], individuals' risk-taking propensity should be aligned with their attitudes toward different health-related behaviors. Intrinsic beliefs of the benefits of vaccines could motivate people to vaccinate.

Regarding the participants' attitudes toward the government, individuals who trusted information from the government and those who agreed with the government's policy of movement tracking and providing financial aid were more willing to vaccinate. In similarity to previous studies [24, 42, 58], trust in government also plays an important role in the vaccination decision-making process. Further, the more value people placed on individual privacy,

the higher their willingness to vaccinate. This effect was not investigated by previous studies. Moreover, according to [3, 59, 60], misinformation is more strongly associated with declines in vaccination intent, and susceptibility to misinformation and vaccine hesitancy lead to a reduced likelihood to comply with health guidance measures. In our study, those who were worried about the negative impact of misinformation also tended to have a higher willingness to vaccinate.

Rather than basing the intention to vaccinate on the threat of infection and its consequences, vaccine intentions are rather based on perceived individual risks such as severity and susceptibility [3, 11, 57]. As mentioned, most of the confirmed cases of the outbreak occurred in Taipei, however, there is no significant difference from many variables for people living in Taipei to have a higher intent to vaccinate. Since most locations in Taiwan can be reached within a few hours, it is reasonable for people from different locations share the same level of perceived risks and vaccination intent. Nevertheless, those who had tested for the COVID-19 virus tended to have higher vaccination intentions. There was no widespread COVID-19 screening before or during the outbreak, however, certain places and professions required recent proof of a COVID-19 negative test result. Therefore those who had had COVID-19 tests, all of which were negative, could have had a higher need to vaccinate, when compared to the general public. Furthermore, the timing and the development of viruses and diseases influenced perceived risks too. In our model, both the timing of the COVID-19 level three alert and the previous daily new confirmed cases had strong impacts on one's vaccination intentions. Our results showed that participants' vaccination intent was highly sensitive to time and the risk of infection people perceive. Accordingly, the vaccination acceptance rate had been growing from 53% in October 2020 [43] to a substantially higher amount, 87% of people vaccinated against COVID-19 in June 2022 [49].

Lastly, various studies showed that concern about side effects of vaccines are the most common cause for hesitancy [6, 17, 23, 27, 30, 35, 37, 41, 61]. In our study, while considering the risks of a vaccination, people's motivation for vaccinating was strengthened by considering the physical impact of the vaccine. This physical impact in our model was neutral, and it was an aspect participants considered. In contrast, the risk of the vaccination process or other reasons lowered their motivation.

This study had its limitations. First, we attempted to use different mediums to recruit participants, but the online survey was biased toward internet users. Additionally, participant panels can be subject to bias and may not be representative of the general population. Our participants tended to have greater internet access and higher socioeconomic status [62]. Secondly, for each participant, this survey only reflected a snapshot taken at a certain point in time, not accommodating for the potential change in a participants' willingness to vaccinate over time [63]. Finally, the study was performed in Taiwan only, limiting our accountability for unique aspects from other countries.

## Conclusions

Using a 10 week nationwide online survey performed during a time period containing the time ahead, the start, and the peak of a COVID-19 outbreak, we investigated many aspects that could affect participants' vaccination intentions. The domains we explored were multidimensional, including individuals' demographic factors, personality traits, QoL, public health knowledge, personal health behavior, attitude toward the government, reasons to consider being vaccinated and COVID-19 vaccination related risk factors. Our study confirmed many findings from previous studies which suggest that being male, older people, improved knowledge and personal health behavior, trust in government, and concern about misinformation

tended to increase vaccination intent. There were some distinctive findings in our study as well. We found that people with an open personality and better QoL in the physical health domain were inclined to vaccinate. Perceived risks played a crucial role in the vaccine decision-making process too. When the pandemic became more severe, participants' vaccination intent increased significantly. The findings of the present study could shed light on individuals' vaccination attitude and may provide governments with an empirical and dynamic base to design tailored strategies to reach higher vaccination rates.

## Author Contributions

**Conceptualization:** Hsuan-Wei Lee, Cheng-Han Leng.

**Formal analysis:** Cheng-Han Leng.

**Supervision:** Ta-Chien Chan.

**Writing – original draft:** Hsuan-Wei Lee, Cheng-Han Leng.

**Writing – review & editing:** Ta-Chien Chan.

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
