## [Decision Letter · Decision Letter 0]

30 May 2022

PONE-D-22-05839Determinants of personal vaccination hesitancy before and after the COVID-19 outbreak in TaiwanPLOS ONE

Dear Dr. Authors,

Thank you for submitting your manuscript to PLOS ONE. After careful consideration, we feel that it has merit but does not fully meet PLOS ONE’s publication criteria as it currently stands. Therefore, we invite you to submit a revised version of the manuscript that addresses the points raised during the review process.

We look forward to receiving your revised manuscript.

Kind regards,

Srikanth Umakanthan

Academic Editor

PLOS ONE

Journal Requirements:

2. 

Please provide additional details regarding participant consent. In the ethics statement in the Methods and online submission information, please ensure that you have specified (1) whether consent was informed and (2) what type you obtained (for instance, written or verbal, and if verbal, how it was documented and witnessed). If your study included minors, state whether you obtained consent from parents or guardians. If the need for consent was waived by the ethics committee, please include this information.

This research was funded by the Academia Sinica, Taiwan (AS-IRB-HS07-109104 and 284

AS-IRB-HS07-109090) for the first author.

This research was funded by the Academia Sinica, Taiwan https://www.sinica.edu.tw/en (AS-IRB-HS07-109104 and AS-IRB-HS07-109090) for the first author. The funders had no role in study design, data collection and analysis, decision to publish, or preparation of the manuscript.

Additional Editor Comments:

Dear Authors,

Thanks for submitting your manuscript to PLOS ONE. Your manuscript was reviewed by two esteemed reviewers and the reports are available.

Minor revisions required as provided by the two esteemed reviewers.

Address these issues point-wise and resubmit the revised manuscript highlighting the changes with both track changes and color highlighted text in the manuscript.

Regards,

Srikanth Umakanthan

Reviewers' comments:

Reviewer's Responses to Questions

**Comments to the Author**

1. Is the manuscript technically sound, and do the data support the conclusions?

Reviewer #1: Yes

Reviewer #2: Yes

2. Has the statistical analysis been performed appropriately and rigorously? 

Reviewer #1: Yes

Reviewer #2: Yes

3. Have the authors made all data underlying the findings in their manuscript fully available?

Reviewer #1: Yes

Reviewer #2: Yes

4. Is the manuscript presented in an intelligible fashion and written in standard English?

Reviewer #1: Yes

Reviewer #2: Yes

5. Review Comments to the Author

Reviewer #1: Thanks a lot for the opportunity to review this interesting and timely manuscript.

In this study, Hsuan-Wei Lee et al, investigated COVID-19 vaccine hesitancy in Taiwan and its associated factors.

This is a timely study considering the recent record number of COVID-19 cases in Taiwan.

The study design is robust, and the authors used a cross-sectional national survey before and during the first noticeable increase in the number of COVID-19 cases in Taiwan between March 2021 and May 2021 to assess the attitude of the general population to COVID-19 vaccination with a large sample size.

Overall, the manuscript is well-written, and the findings represent a significant contribution to the field of COVID-19 vaccine hesitancy.

The manuscript is well organized and is comprehensively described and it was written in correct and readable language.

The research design was proper, the methods were adequately described, the results were presented clearly with sufficient Tables, and the conclusions were supported by the results.

Importantly, the potential limitations were presented clearly by the authors.

Minor comments:

1. The authors can benefit from revising the title which can be a bit vague for the readers from countries where the first wave (outbreak) of COVID-19 occurred in the first half of 2020 rather than 2021.

2. In the Introduction section, lines 9-18, the authors can benefit from two additional reference on COVID-19 vaccine acceptance rates worldwide and in Taiwan:

A. https://doi.org/10.2147/JMDH.S347669

B. https://doi.org/10.3390/ijerph18115579

3. In the Introduction section, lines 33-37, the authors can benefit from adding more information regarding the number of cases and mortalities that occurred during the first wave (if it can be called so) of COVID-19 in Taiwan in May 2021

4. In the Introduction section, line 39: please add a reference to support this statement

5. In Figure 1, please add the reference from which this relevant data was retrieved.

6. In Table 1, please add a footnote to spell out the abbreviations (e.g. WHO). The same applies for Table 2 (USD)

7. In the Discussion section, the authors can benefit from adding a paragraph to compare the acceptance rate to COVID-19 vaccination in Taiwan to that in neighbouring countries which were much higher (e.g. China 82%, South Korea 77%) using the aforementioned reference: https://doi.org/10.2147/JMDH.S347669

Reviewer #2: Comments to the Author

My comments are as follows:

1. Abstract- Well structured and summarizes the overall purpose of the study and the research problem(s) investigated. The basic design of the study; major findings and trends found as a result of the study is also showcased.

2. Materials and methods- The authors have included a proper survey questionnaire along with the assessment scale to analyze their objectives.

3. Statistics: The involvement of regression methods (logistic and linear models) provides clearance of bias that may have generated by the different variables. The confounding/impending factors are well neutralized by the statistical methods.

4. Discussion: The discussion requires more additional statements with regards to the existing literature search. Include the following references and citations reflecting the COVID-19 updates to strengthen the manuscript:

- Origin and transmission (use reference and cite:  doi:10.1136/postgradmedj-2020-138234)

- To mention in brief about vaccines (use reference and cite: “doi: 10.1136/postgradmedj-2021-141365.

AND  doi:10.1136/postgradmedj-2021-140654”)

- Definition of vaccine resistance and hesitance (use reference and cite: doi:10.3390/vaccines9101064).

- Compare the global vaccine status and relate it with the current vaccine status (refer and cite:  doi:10.3934/publichealth.2021053)

5. Conclusion: The authors have shown the importance of variables that can influence the adherence of preventive measures and willingness to vaccinate against COVID-19.

The main bulletin messages showcases the summary of the manuscript very well.

I advocate this article for revision pending inclusion of the points as recommended by me.

6. PLOS authors have the option to publish the peer review history of their article (what does this mean?). If published, this will include your full peer review and any attached files.

Reviewer #1: No

Reviewer #2: No

---

## [Author Response · Author response to Decision Letter 0]

6 Jun 2022

Authors’ response

We deeply thank the referees for their insightful and helpful comments on our paper. We believe they have helped us to improve the paper. We have made according revisions to the manuscript. 

In order to meet the requirement of the reviewer, a significant change to the paper is that we rename the title to “Determinants of personal vaccination hesitancy before and after the mid-2021 COVID-19 outbreak in Taiwan.” We believe this title is more consistent with the content of the manuscript too.

Moreover, in order to meet the requirements from the journal, we made some cosmetic changes of ethic statements in the new manuscript too. 

Below we respond to individual comments from the referees. We use black text for the reviewers’ comments and our reply in blue. In the manuscript, we use blue color for the text we added. 

Reviewer #1: 

Thanks a lot for the opportunity to review this interesting and timely manuscript. In this study, Hsuan-Wei Lee et al, investigated COVID-19 vaccine hesitancy in Taiwan and its associated factors. This is a timely study considering the recent record number of COVID-19 cases in Taiwan.

The study design is robust, and the authors used a cross-sectional national survey before and during the first noticeable increase in the number of COVID-19 cases in Taiwan between March 2021 and May 2021 to assess the attitude of the general population to COVID-19 vaccination with a large sample size. Overall, the manuscript is well-written, and the findings represent a significant contribution to the field of COVID-19 vaccine hesitancy. The manuscript is well organized and is comprehensively described and it was written in correct and readable language. The research design was proper, the methods were adequately described, the results were presented clearly with sufficient Tables, and the conclusions were supported by the results. Importantly, the potential limitations were presented clearly by the authors.

Minor comments:

1. The authors can benefit from revising the title which can be a bit vague for the readers from countries where the first wave (outbreak) of COVID-19 occurred in the first half of 2020 rather than 2021.

We agree with the reviewer that the old title is a little bit unclear. Following the reviewer’s suggestion, we change the title of this article as “Determinants of personal vaccination hesitancy before and after the mid-2021 COVID-19 outbreak in Taiwan.” We deeply appreciate the reviewer’s valuable suggestion. 

2. In the Introduction section, lines 9-18, the authors can benefit from two additional reference on COVID-19 vaccine acceptance rates worldwide and in Taiwan:

A. https://doi.org/10.2147/JMDH.S347669

B. https://doi.org/10.3390/ijerph18115579

We thank the reviewer’s suggestion. The two additional references are indeed relevant to this research and they are both added in the new manuscript (line 14). We also added two supplementary references to illustrate the initial success of controlling the COVID-19 epidemic (line 27). 

3. In the Introduction section, lines 33-37, the authors can benefit from adding more information regarding the number of cases and mortalities that occurred during the first wave (if it can be called so) of COVID-19 in Taiwan in May 2021.

We appreciate the reviewer’s suggestion. The information regarding the number of cases and mortalities are now provided (line 36-40). 

4. In the Introduction section, line 39: please add a reference to support this statement.

We thank the reviewer’s suggestion and add a reference to support the statement in the new manuscript (line 45). 

5. In Figure 1, please add the reference from which this relevant data was retrieved. 

We thank the reviewer’s suggestion. The dataset we referred to is cited in the new manuscript (line 64). 

6. In Table 1, please add a footnote to spell out the abbreviations (e.g. WHO). The same applies for Table 2 (USD) 

We appreciate the reviewer’s reminder and spell out the two abbreviations we used in the new manuscript. 

7. In the Discussion section, the authors can benefit from adding a paragraph to compare the acceptance rate to COVID-19 vaccination in Taiwan to that in neighbouring countries which were much higher (e.g. China 82%, South Korea 77%) using the aforementioned reference: https://doi.org/10.2147/JMDH.S347669

We thank the reviewer’s suggestion and think it is indeed important for us to make a comparison of vaccination acceptance rates to the nearby countries. However, the reason Taiwan has a relatively low acceptance rate to COVID-19 vaccination in this study is mainly because that the survey was done in the period that there was no local COVID-19 cases spreading in Taiwan for several months. When the pandemic is eminent, people’s vaccination acceptance rates would grow much higher, and this is also consistent to our main finding in the manuscript. Our results showed that participants’ vaccination intent was highly sensitive to time and the risk of infection people perceive. To put elaborate on this more clearly, we added several sentences in the new manuscript accordingly (line 28-30 and 259-262). 

Reviewer #2: Comments to the Author

My comments are as follows:

1. Abstract- Well structured and summarizes the overall purpose of the study and the research problem(s) investigated. The basic design of the study; major findings and trends found as a result of the study is also showcased.

2. Materials and methods- The authors have included a proper survey questionnaire along with the assessment scale to analyze their objectives.

3. Statistics: The involvement of regression methods (logistic and linear models) provides clearance of bias that may have generated by the different variables. The confounding/impending factors are well neutralized by the statistical methods.

4. Discussion: The discussion requires more additional statements with regards to the existing literature search. Include the following references and citations reflecting the COVID-19 updates to strengthen the manuscript:

- Origin and transmission (use reference and cite: doi:10.1136/postgradmedj-2020-138234)

- To mention in brief about vaccines (use reference and cite: “doi: 10.1136/postgradmedj-2021-141365. AND doi:10.1136/postgradmedj-2021-140654”)

- Definition of vaccine resistance and hesitance (use reference and cite: doi:10.3390/vaccines9101064).

- Compare the global vaccine status and relate it with the current vaccine status (refer and cite: doi:10.3934/publichealth.2021053)

5. Conclusion: The authors have shown the importance of variables that can influence the adherence of preventive measures and willingness to vaccinate against COVID-19.

The main bulletin messages showcases the summary of the manuscript very well.

I advocate this article for revision pending inclusion of the points as recommended by me.

We thank the reviewer for the nice comments. Moreover, we appreciate the reviewer’s suggestion and we think these references could make our research more comprehensive. The above references are indeed related to this research and we add these references in the new manuscript (in line 2, line 10, line 264, line 11, and line 13, respectively).

---

## [Decision Letter · Decision Letter 1]

9 Jun 2022

Determinants of personal vaccination hesitancy before and after the mid-2021 COVID-19 outbreak in Taiwan

PONE-D-22-05839R1

Dear Dr. Lee,

We’re pleased to inform you that your manuscript has been judged scientifically suitable for publication and will be formally accepted for publication once it meets all outstanding technical requirements.

Kind regards,

Srikanth Umakanthan

Academic Editor

PLOS ONE

Additional Editor Comments (optional):

Reviewers' comments:

Reviewer's Responses to Questions

**Comments to the Author**

1. If the authors have adequately addressed your comments raised in a previous round of review and you feel that this manuscript is now acceptable for publication, you may indicate that here to bypass the “Comments to the Author” section, enter your conflict of interest statement in the “Confidential to Editor” section, and submit your "Accept" recommendation.

Reviewer #1: All comments have been addressed

Reviewer #2: All comments have been addressed

2. Is the manuscript technically sound, and do the data support the conclusions?

Reviewer #1: Yes

Reviewer #2: Yes

3. Has the statistical analysis been performed appropriately and rigorously? 

Reviewer #1: Yes

Reviewer #2: Yes

4. Have the authors made all data underlying the findings in their manuscript fully available?

Reviewer #1: Yes

Reviewer #2: Yes

5. Is the manuscript presented in an intelligible fashion and written in standard English?

Reviewer #1: Yes

Reviewer #2: Yes

6. Review Comments to the Author

Reviewer #1: Thanks for addressing all the previous comments properly and thoroughly.

The manuscript is well written and provides timely and important results.

Reviewer #2: The authors have addressed all of my comments and the manuscript now appears to be of sound nature and of acceptable standards in accordance with the journal requirements.

7. PLOS authors have the option to publish the peer review history of their article (what does this mean?). If published, this will include your full peer review and any attached files.

Reviewer #1: No

Reviewer #2: No

---

## [Editor Report · Acceptance letter]

13 Jun 2022

PONE-D-22-05839R1 

Determinants of personal vaccination hesitancy before and after the mid-2021 COVID-19 outbreak in Taiwan 

Dear Dr. Lee:

I'm pleased to inform you that your manuscript has been deemed suitable for publication in PLOS ONE. Congratulations! Your manuscript is now with our production department. 

Kind regards, 

on behalf of

Dr. Srikanth Umakanthan 

Academic Editor

PLOS ONE